# Efficacy of Quantitative Pupillary Light Reflex for Predicting Neurological Outcomes in Patients Treated with Targeted Temperature Management after Cardiac Arrest: A Systematic Review and Meta-Analysis

**DOI:** 10.3390/medicina58060804

**Published:** 2022-06-15

**Authors:** Jae-Guk Kim, Hyungoo Shin, Tae-Ho Lim, Wonhee Kim, Youngsuk Cho, Bo-Hyoung Jang, Kyu-Sun Choi, Min-Kyun Na, Chiwon Ahn, Juncheol Lee

**Affiliations:** 1Department of Emergency Medicine, College of Medicine, Hallym University, Chuncheon 24253, Korea; gallion00@gmail.com (J.-G.K.); wonsee02@gmail.com (W.K.); faith2love@hanmail.net (Y.C.); 2Department of Emergency Medicine, Hanyang University College of Medicine, Seoul 04763, Korea; seodtst@gmail.com (H.S.); jclee0221@gmail.com (J.L.); 3Department of Preventive Medicine, College of Korean Medicine, Kyung Hee University, Seoul 02447, Korea; bhjang@khu.ac.kr; 4Department of Neurosurgery, Hanyang University College of Medicine, Seoul 04763, Korea; vertex-09@hanmail.net (K.-S.C.); mavmav@hanmail.net (M.-K.N.); 5Department of Emergency Medicine, College of Medicine, Chung-Ang University, Seoul 06974, Korea; cahn@cau.ac.kr

**Keywords:** pupillometry, targeted temperature management, heart arrest, patient outcome assessment, me-ta-analysis

## Abstract

*Background and objectives*: This study aims to evaluate the usefulness of the quantitative pupillary light reflex as a prognostic tool for neurological outcomes in post-cardiac arrest patients treated with targeted temperature management (TTM). *Material and Methods*: We systematically searched MEDLINE, EMBASE, and the Cochrane Library (search date: 9 July 2021) for studies on post-cardiac arrest patients treated with TTM that had measured the percent constriction of pupillary light reflex (%PLR) with quantitative pupillometry as well as assessed the neurological outcome. For an assessment of the methodological quality of the included studies, two authors utilized the prognosis study tool independently. *Results*: A total of 618 patients from four studies were included in this study. Standardized mean differences (SMDs) were calculated to compare patients with good or poor neurological outcomes. A higher %PLR measured at 0–24 h after hospital admission was related to good neurological outcomes at 3 months in post-cardiac arrest patients treated with TTM (SMD 0.87; 95% confidence interval 0.70–1.05; I^2^ = 0%). A higher %PLR amplitude measured at 24–48 h after hospital admission was also associated with a good neurological outcome at 3 months in post-cardiac arrest patients treated with TTM, but with high heterogeneity (standardized mean difference 0.86; 95% confidence interval 0.40–1.32; I^2^ = 70%). The evidence supporting these findings was of poor quality. For poor neurological outcome, the prognosis accuracy of %PLR was 9.19 (pooled diagnostic odds ratio, I^2^ = 0%) and 0.75 (area under the curve). *Conclusions*: The present meta-analysis could not reveal that change of %PLR was an effective tool in predicting neurological outcomes for post-cardiac arrest patients treated with TTM owing to a paucity of included studies and the poor quality of the evidence.

## 1. Introduction

Sudden cardiac arrest is a major public health challenge correlated with a high rate of poor outcomes [1]. Although advances in medical treatment and resuscitation care, such as percutaneous coronary intervention and targeted temperature management (TTM), have improved the outcomes of post-cardiac arrest patients [2,3], only a small portion of hospitalized cardiac arrest survivors are discharged with a good neurological outcome (GNO) [4]. Several prognostic methods, such as neurological exams, somatosensory evoked potential, electroencephalography (EEG), and serum markers such as neuron-specific enolase, have therefore been investigated. However, predicting neurological prognosis after TTM following cardiac arrest remains difficult [1].

The evaluation of pupillary light reflex (PLR) is an essential element of neurological assessment. For example, a bilaterally absent PLR at 72 h or more after cardiac arrest is an indicator of poor outcome irrespective of TTM treatment [1,2,3]. However, the traditional qualitative assessment of PLR has poor inter-rater reliability as it involves an evaluator using a penlight and describing the response as absent, sluggish, or brisk [4,5,6].

The PLR was recently quantified using an automated pupilometer (AP), which provides an objective measurement of pupil size and how it changes in response to light stimulus, to calculate the percent constriction of the pupillary light reflex (%PLR) in hypoxic-ischemic encephalopathy following cardiac arrest [7,8].

Previous studies found that a higher %PLR was associated with GNO in post-cardiac arrest patients treated with TTM [7,9,10,11]. However, the prognostic value of the quantitative pupilometer for these patients varies according to measurement time among studies. Therefore, we conducted a systematic review and meta-analysis to investigate the predictive value of the quantitative measurement of PLR using an AP for neurological prognosis in post-cardiac arrest patients who were treated with TTM.

## 2. Materials and Methods

We adhered to the Meta-analysis of Observational Studies in Epidemiology [12] and Preferred Reporting Items for Systematic Reviews and Meta-analysis (PRISMA) reporting guidelines [13]. The protocol for the review was registered at http://www.crd.york.ac.uk/PROSPERO/ (registration number: CRD42020136980) accessed on 24 March 2020.

We formulated a PICO question based on population (P), intervention (I), comparison (C), and outcome (O). The following was the PICO question: population (P) = adult post-cardiac arrest patients treated with TTM; intervention (I) = %PLR as measured by an AP; comparator (C) = none; outcome (O) = GNO at and after hospital discharge.

### 2.1. Search Strategy

We conducted a database search to evaluate the predictive value of a quantitative pupilometer in cardiac arrest patients treated with TTM. Two experienced reviewers (J.G.K and H.S) performed a study search on 9 July 2021. MEDLINE and EMBASE databases were used to search (MEDLINE: from January 1975 to May 2020; EMBASE: from January 1975 to October 2020) via the Ovid interface, and the Cochrane Library (all years) was searched for relevant articles with no language restrictions. In addition, we manually checked the references of the qualified research to identify other relevant studies.

The search terms included: “cardiac arrest” or “cardiopulmonary resuscitation” or “CPR” or “return of spontaneous resuscitation” or “ROSC” or “advanced cardiac life support” and “therapeutic hypothermia” or “targeted temperature management” or “TTM” and “pupillary light reflex” or “pupilometer” or “pupillometry” (Appendix A). We included all articles that addressed our PICO question.

### 2.2. Automated Pupillometer

Quantitative pupillometry is the objective measurement of pupil size and reactivity using a portable, automatic device that emits a standard light-emitting diode light source and records pupil responsiveness. Two types of quantitative AP devices were used to record %PLR in the included studies. A NPi-200 pupillometer (NeurOptics, Irvine, CA, USA) was used in two studies [7,10], and a NeuroLight Algiscan (IDMED, Marseille, France) was used in the other two studies [9,11].

### 2.3. Study Selection

Two reviewers (J.-G.K. and H.S.), based on the article’s title, abstract, and study design, independently selected all studies according to the predetermined selection criteria. Following are the exclusion criteria for this review: irrelevant populations (cardiac arrest patients without a sustained return of spontaneous resuscitation, patients not treated with TTM, pediatric populations, and animal studies), irrelevant data (studies not including pupillometry data), case reports, reviews, editorials, and duplicate data. We also removed duplicate publications by comparing the titles, authors, and publication dates of all identified studies. In the case of a disagreement between the two reviewers (J.-G.K and H.S.), a third reviewer (T.-H.L.) intervened, and differences were discussed until a consensus for the agreement was attained. After eliminating the excluded abstracts, we rescreened the full text of the selected articles and evaluated them more thoroughly for eligibility using the same inclusion and exclusion criteria.

We described the reasons why potentially relevant studies did not match the eligibility criteria (Appendix A), and the search results are presented in a PRISMA flowchart (Figure 1). After excluding ineligible studies, we retrieved the entire texts of the remaining studies and reevaluated them using the same criteria.

### 2.4. Data Extraction

Data extraction from the included studies was completed separately by two reviewers (J.G.K. and H.S.). Any disagreement between reviewers that remained unresolved was discussed and re-resolved by consensus or reviewed by a third reviewer (T.-H.L.). The variables extracted were as follows: the author, year of publication, study design, country, inclusion period, sample size, cardiac arrest type (out-of-hospital cardiac arrest [OHCA] or in-hospital cardiac arrest [IHCA]), neurological outcome, equipment used for %PLR measurement, %PLR assessment time (at 0–24 h vs. 24–48 h), neurological assessment tool (Cerebral Performance Category Scale; CPC), time points of outcome measurement (at hospital discharge and 3 months), and mean (±standard deviation (SD)) %PLR level. If estimated mean (SD) data were unavailable, median values with interquartile ranges or median values with lowest and maximum ranges were used to derive estimated mean (SD) values using the approach of Wan et al. [14]. The neurological outcome was classified as good or poor based on CPC scores (1–2: good neurological outcome; 3–5: poor neurological outcome).

To determine the prognostic accuracy of %PLR in predicting poor neurological outcome (PNO) in post-cardiac arrest patients treated with TTM. We generated two-by-two tables with two variables for %PLR and neurological outcome. In addition, each study’s true-positive (TP), false-positive (FP), false-negative (FN), and true-negative (TN) data were obtained. The two-by-two tables offered in the included studies were as follows: TP = PNO with decreased %PLR, FP = GNO with decreased %PLR, FN = PNO with an increase in %PLR, and TN = GNO with an increase in %PLR. If none of these variables were specified in the studies, we emailed the corresponding authors with pertinent queries for additional information.

### 2.5. Risk of Bias in Individual Studies

The methodological quality of the four included studies were evaluated separately by two reviewers (J.-G.K. and H.S.) blinded to authorship and the journal using the Quality in Prognosis Studies (QUIPS) tool, with values of 2, 1, and 0 indicating low, unclear, and high risk, respectively [15]. A study was rated high quality if it received 9 or more points in the sum of each 6-item assessment. Unresolved disagreements among reviewers was settled through discussion or review by a third author (T.-H.L.).

### 2.6. Statistical Analysis

In the main analysis, the association between %PLR amplitude and neurological outcomes was assessed by the standardized mean differences (SMDs) between the patients with GNO and the patients with PNO using a random-effects model. The %PLR across comparison groups was extracted as mean differences with 95% confidence intervals (CIs).

For the measurement of heterogeneity, I^2^ statistics were employed to quantify the proportion of study inconsistency, with values of 25%, 50%, and 75%, respectively, regarded to be low, moderate, and high [16].

In addition, we intended to establish the predictive accuracy of %PLR in predicting PNO in patients treated with TTM following cardiac arrest. Pooled sensitivities, specificities, positive and negative likelihood ratios, and diagnostic odds ratios (DOR) were calculated by generating two-by-two tables with two variables for %PLR and neurological outcome. Summary receiver-operating characteristic curves (SROC) were constructed and used to characterize the overall prognostic performance of the percent PLR, which represented the calculated Q* index and area under the characteristic curve (AUC) [17,18]. AUC values were categorized as follows: greater than 0.97 (excellent), 0.93 to 0.96 (very good), 0.75 to 0.92 (good), and less than 0.75 (reasonable but obviously deficient in prognostic accuracy) [19].

The Endnote X9 (Clarivate Analytics, Philadelphia, PA, USA) reference management software was utilized to arrange all identified studies from the literature search. We performed statistical analyses using Meta-Disc, version 1.4 (Clinical Biostatistics, Ramony Cajal Hospital, Madrid, Spain), and RevMan, version 5.3 (Cochrane Collaboration, Oxford, UK), and *p* < 0.05 was considered statistically significant. Using the R package “meta”, sensitivity analysis and publication bias analysis was also conducted (R, version 3.3.2; R Foundation for Statistical Computing, Vienna, Austria). Sensitivity analysis was done by sequentially omitting individual studies. Publication bias was assessed by funnel plot and Egger’s test. The asymmetry of the funnel plot and *p*-value (<0.05) using Egger’s test indicated that bias existed.

The GRADEpro Guideline Development Tool (McMaster University and Evidence Prime, Inc., Hamilton, ON, Canada) was utilized to assess the quality of evidence in each study. The evidence was summarized based on GRADE levels (high, moderate, low, and very low) by analyzing the included studies’ designs, risk of bias, consistency, precision, directness, and potential publication bias.

### 2.7. Outcome Measures

The primary outcomes were GNO at hospital discharge and 3 months later. According to the CPC scores, the neurological outcome was classified as good (CPC 1–2) or poor (CPC 3–5).

## 3. Results

### 3.1. Study Selection and Characteristics of Included Studies

Through database scanning, 268 records were identified (Figure 1). The titles and abstracts of 184 records were evaluated for eligibility after deleting 84 duplicates. Fifty records were selected as potentially pertinent, and full-text articles were retrieved for a more in-depth analysis. Four studies including 618 patients were included in the meta-analysis following the exclusion of 46 articles based on the full-text evaluation. Among them, 247 (40.0%) patients had a GNO and 494 (80%) patients were male.

Table 1 describes the main characteristics of the four eligible studies. In addition, baseline patient characteristics are presented in Appendix A. TTM related information including target temperature, maintenance time, rewarming rate, TTM devices, sedative drugs, and other information such as conflict of interest and blinding are summarized in Appendix A.

Four studies were prospective observational studies. Among them, three studies were single-center studies conducted in Europe and the USA and the other study was a multi-center study conducted in Europe. Three studies included both OHCA and IHCA patients, while one study exclusively involved OHCA patients. The %PLR was measured at 0–24 h in four studies, at 24–48 h in three studies, and 48–72 h in one study. All four studies evaluated neurological outcomes using the CPC and deemed a CPC score between 3 and 5 to indicate a poor neurological outcome. The measurement time of neurological outcome was at hospital discharge or 3 months later (Table 1).

### 3.2. Quality of the Included Studies

The quality grading system was utilized to evaluate the four included studies, with one study assessed as low quality [7] and the other three studies evaluated as high quality [9,10,11]. Appendix A provide a summary of our evaluation of the risk of bias in the included studies.

### 3.3. Main Analysis

Four relevant studies with a total of 618 patients were analyzed. Four studies reported differences between the GNO and PNO groups in the %PLR.

In the meta-analysis, the %PLR measured at 0–24 h from hospital admission was significantly elevated in the patients with GNO compared with the patients with PNO, demonstrating a positive association with an overall SMD [(mean value in the GNO group–mean value in the PNO group)/pooled SD] of 0.81 (95% CI, 0.60–1.02; I^2^ = 16%; *p* < 0.00001, Figure 2). In the analysis for predicting neurological outcome at 3 months, the %PLR level in the patients with GNO was relatively higher than that in the patients with PNO (three studies; SMD, 0.87; 95% CI, 0.70–1.05; I^2^ = 0%; *p* < 0.00001).

The %PLR measured at 24–48 h from hospital admission was also significantly higher in the patients with GNO than in the patients with PNO. In the analysis of time points for measuring neurological outcome at 3 months, the %PLR in the patients with GNO was relatively higher than in the patients with PNO but there was significant heterogeneity (three studies; SMD, 0.86; 95% CI, 0.40–1.32; I^2^ = 70%; *p* < 0.00002; Figure 3).

### 3.4. Sensitivity Analysis

Substantial heterogeneity (I^2^ = 70%) was present in the meta-analysis of %PLR measured at 24–48 h for predicting neurological outcomes at 3 months. In the sensitivity analysis, we detected one study by Suys et al. which showed significant heterogeneity (Appendix A). After sensitivity analysis and exclusion of the outlier, I^2^ statistics decreased to 56% and the %PLR in the patients with GNO was relatively higher than that of the patients with PNO (two studies; SMD, 1.04; 95% CI, 0.63–1.46; I^2^ = 56%; *p* = 0.00001; Appendix A).

### 3.5. Prognostic Accuracy of Percent Constriction of Pupillary Light Reflex in Predicting Poor Neurological Outcome

Three studies were included in summary estimates with the prognostic accuracy of %PLR in predicting poor neurological outcome [7,9,11]. All studies were single-center studies. %PLR was measured in patients with OHCA and IHCA in one study [11] and OHCA in two studies [7,9]. The proportion of patients with PNO was 64.7% of the overall patient population of 187. The time point for measuring neurological outcomes varied from hospital discharge to 3 months.

The pooled DOR value of %PLR for predicting PNO was 9.19 (95% CI = 4.03–20.93: I^2^ = 0%). The AUC value was 0.75 (standard error = 0.09; Q = 0.69) in three studies (Figure 4). The sensitivity and specificity of %PLR ranged from 0.45 to 0.66 and 0.78 to 0.91, respectively (Appendix A). The pooled sensitivity and specificity were 0.58 (95% CI = 0.49–0.67; I^2^ = 52.8%) and 0.86 (95% CI = 0.76–0.94; I^2^ = 31.1%), respectively. The pooled positive and negative likelihood ratios were 3.94 (95% CI = 2.05–7.57; I^2^ = 5.9) and 0.49 (95% CI = 0.37–0.65; I^2^ = 29.1), respectively (Appendix A).

### 3.6. Publication Bias and Quality of Evidence According to GRADE Levels

In the funnel plot, there was no clear asymmetry. Based on Egger’s regression test, the assessment had no significant bias (at 0–24 h, *p* = 0.4553; at 24–48 h, *p* = 0.9579; Figure 5). The included study had a low quality of evidence for good neurological outcome at 3 months in both time points (at 0–24 h and 24–48 h, Appendix A).

## 4. Discussion

In post-cardiac arrest patients treated with TTM, we founded that a higher %PLR measured within 24 h after hospital admission is related with GNO at 3 months. and the summary estimates of the sensitivity and specificity of %PLR in predicting PNO were 0.58 (95% CI = 0.49–0.67; I^2^ = 52.8%) and 0.86 (95% CI = 0.76–0.94; I^2^ = 31.1%), respectively. The pooled DOR value of %PLR in post-cardiac arrest patients treated with TTM was 9.19 (95% CI = 4.03–20.93; I^2^ = 0%), and the AUC value was 0.75. However, it is difficult to determine if the change in %PLR is relevant for predicting neurological prognosis in post-cardiac arrest patients treated with TTM. This is because we discovered that the results varied based on the measurement time of %PLR and that it was unable to conclude due to a paucity of included studies and evidence of low quality according to the GRADE level. In addition, the test’s practical relevance is limited by the insufficient sensitivity and specificity of %PLR for predicting neurological prognosis.

Several prognostic methods, such as EEG, somatosensory evoked potential (SSEP), neuron-specific enolase, diffuse-weighted magnetic resonance imaging, grey matter to white matter ratio in brain computed tomography, and optic nerve sheath diameter, have been used to predict the prognosis for post-cardiac arrest patients [20]. However, These techniques are not always available, and interpreting EEG and SSEP requires a high level of expertise.

During the neurological recovery of post-cardiac arrest, patients had favorable neurological outcomes following ROSC, brainstem reflex function returns faster than both cortical function and consciousness [3,21]. Physiologically, pupil size is controlled by the hypothalamus-centered autonomic nerve system [22]. The hypothalamus is a crucial component of the ascending activation system that regulates arousal [23]. Consequently, the PLR monitors not only the brainstem reflex but also more complex responses, including hypothalamic functions such as autonomic nervous system responses [10,24]. Therefore, based on the measurement of PLR, a physician can predict a better neurological prognosis. However, for patients treated with TTM, the PLR may not always be a reliable indicator of neurological recovery [7,8,14] because sedative drugs and TTM can affect pupil size [5,6].

The detection of the presence of a PLR is impossible when the reduction in the PLR is <0.3 mm [9,11,25]. In addition to the damage to the midbrain or other regions of the brain, reduced pupil diameters may be caused by a variety of other circumstances, such as advanced age, general anesthesia, opiates, diabetic neuropathy, cataracts, and others [26]. AP is more reliable than standard clinical pupillary assessments, especially if pupils are small [13,14,15], is portable, needs a short time (<30 s) for measurement, is not operator-dependent, and can measure the degree of the PLR [27].

Several quantitative pupillary parameters can be measured using an AP, including maximum diameter, minimum diameter, constriction velocity (CV), maximum constriction velocity (MCV), dilation velocity (DV), and neurological pupil index (NPi) [28].

A recent study reported that NPi showed high specificity in predicting PNO in a large multicenter study [10]. However, NPi was not available from any APs because NPi can only be calculated by the NPi-200 pupillometer. Among the included studies in this analysis, only Oddo et al. and Riker et al. reported NPi in their studies (Appendix A). %PLR can be measured from both APs used in the included studies and is clinically familiar to a physician [29]. One recent study showed that PLR was favorably linked with various quantitative pupillary response indices, including CV, MCV, DV, and NPi [29]. The NPi calculation consists of several pupillary measures with a considerable correlation to %PLR. Taken together, NPi and %PLR are moderately correlated [29], and thus %PLR could be a valuable parameter for predicting GNO when using an AP.

Several studies have shown the prognostic value of %PLR by AP for post-cardiac arrest patients treated with TTM. Solari et al. showed that the %PLR predicted poor outcomes better than pupil size, EEG, and SSEP [30]. Furthermore, they found that the pupillometer was superior to manual pupil assessment, and their quantitative %PLR was higher in patients with a GNO [11].

Importantly, previous studies measured the %PLR at different times to determine the best time for the prediction of neurological outcomes. In the included studies, %PLR was measured at 0–24, 24–48, and 48–64 h after hospital admission. The %PLR was consistently greater in survivors with favorable neurological outcomes. However, the prognostic performance of the %PLR was inconsistent across various time points, and the optimal timing for prognostication remains unclear.

In this meta-analysis for %PLR and neurological outcome, patients with a relatively higher %PLR within 24 h after hospital admission showed a significantly better association with GNO than those with a lower %PLR. However, only three observational studies were included for this outcome, and the GRADE level for the quality of the evidence supporting this result was low. The %PLR measured at 24–48 h after hospital admission showed a significantly better association with GNO than a lower %PLR. However, this result had high heterogeneity (I^2^ = 70%), indicating a less reliable association with neurological outcomes. The study by Suys et al. was considered as a major contribution to the high heterogeneity because this study was conducted only on OHCA patients [11]. In general, the neurological outcome of OHCA patients was worse than those of IHCA patients. We assumed that the different patient features of this study would result in a significant degree of heterogeneity. To resolve this potential high heterogeneity, a sensitivity analysis was performed for the included studies, but the heterogeneity was not resolved (I^2^ = 56%). The GRADE level for this result was also low.

Concerning the predictive accuracy of %PLR for neurological outcomes, the study showed a significant pooled DOR with low heterogeneity. (DOR = 9.19 (95% CI = 4.03–20.93: I^2^ = 0%). However, the sensitivity and specificity of %PLR for predicting neurological prognosis were insufficient

Recent editorials have demonstrated that quantitative approaches utilizing pupilometer devices for pupillary reflex to predict neurological outcome in post-cardiac arrest patients are insensitive and have a weak correlation with outcome prediction [31]. Higher brain areas, including the thalamus and cerebral cortex, are both essential for con-sciousness and more vulnerable to hypoxia than the brainstem. Furthermore, even in the absence of hypoxic-ischemic brain injury, patients with abnormal peripheral nerve con-duction or blunted sympathetic response may reduce pupil reflex, and accessory spinal pathways that synapse to the cervical sympathetic ganglia can cause pupillary dilation in response to pain after brain death [31,32]. This intricacy of pupil reflex measurement could lead to inconsistency in predicting the neurological prognosis of patients treated with TTM following cardiac arrest. Consequently, it is fair to evaluate AP within the framework of an established, multimodal neuroprognostic methodology.

This meta-analysis has several limitations. First, the study result cannot be generalized to other populations because all the included studies were confined to Europe and the USA and only one study was a multicenter study. For a more generalizable conclusion, further studies including data from other races or countries are required. Second, the method of measurement of the %PLR in the eyes and the sedation level of patients could be confounding factors. For example, Heimberger et al. reported the mean value of %PLR for both eyes [9] and Riker et al. used the worst value measured in both eyes at each timepoint [7], while the other researchers did not describe how they dealt with differences in the eyes [7,10]. Another possible confounding factor was the level of sedation of the patients, as the degree of sedation was not formally assessed using a standardized sedation scale. The sedative drugs used, and the depth of sedation was different in each of the included studies. Therefore, the difference in measurement methods for assessing the %PLR in both eyes as well as the sedation level of patients in this study may have contributed to the degree of %PLR and should be considered when interpreting the reported outcome of this study. Third, CPC measured at 3 months after hospital discharge can be changed up until 6 months, suggesting that the results could have been different if a long-term CPC test was taken after 6 months [20].

## 5. Conclusions

The present meta-analysis could not reveal that change of %PLR was an effective tool in predicting neurological outcomes for post-cardiac arrest patients treated with TTM due to a lack of included studies and low quality of evidence. For a more robust predictive value of %PLR, more studies that assessed longer-term outcomes and have a large sample size of patients are required and specific analyzes should be conducted on the influence of sedation and temperature achieved during TTM on the measurements and outcome prediction.

## Figures and Tables

**Figure 1 medicina-58-00804-f001:**
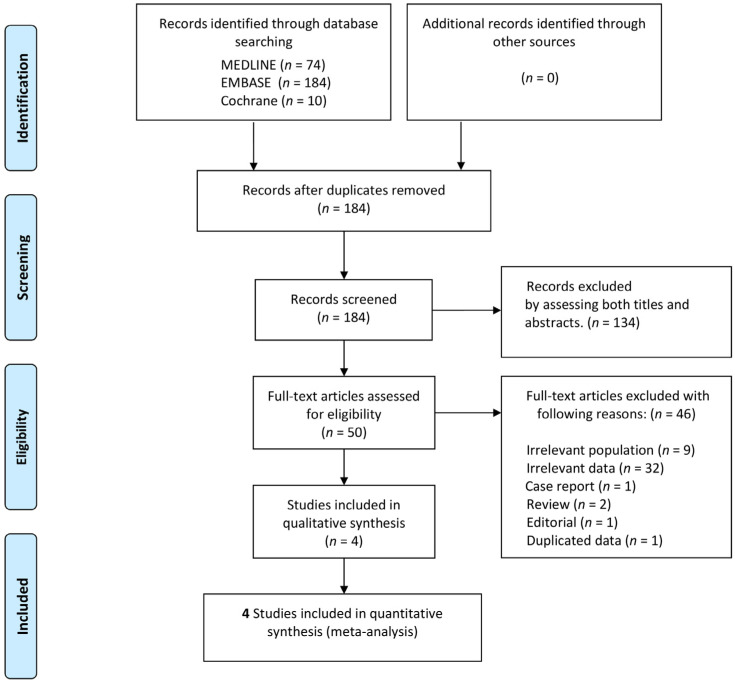
The flow diagram depicts the studies that were included in the meta-analysis.

**Figure 2 medicina-58-00804-f002:**
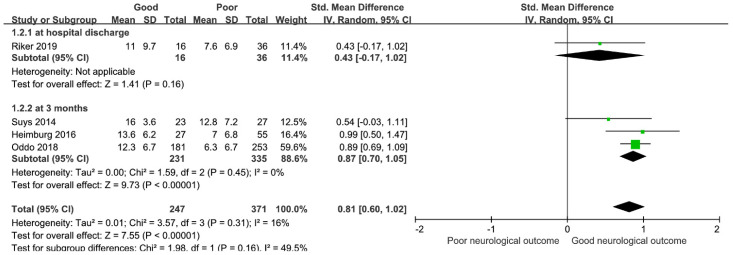
Forest plot for percent constriction of pupillary light reflex measured at 0–24 h after hospital admission and neurological outcome at hospital discharge and 3 months. CI, confidence interval; SD, standard deviation.

**Figure 3 medicina-58-00804-f003:**
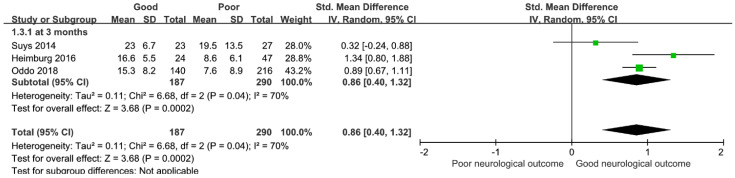
Forest plot for percent constriction of pupillary light reflex measured at 24–48 h after hospital admission and neurological outcome at 3 months. CI, confidence interval; SD, standard deviation.

**Figure 4 medicina-58-00804-f004:**
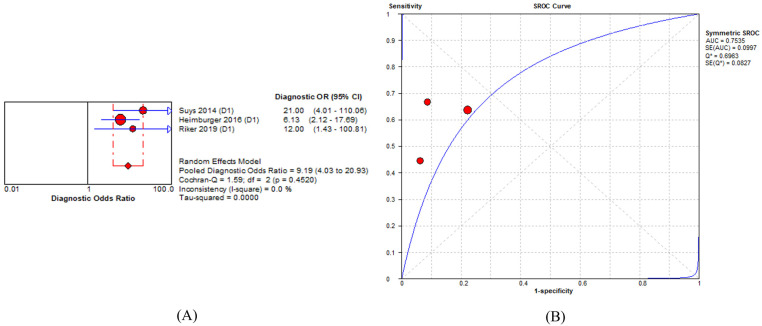
Prognostic accuracy of percent constriction of pupillary light reflex for a poor neurological outcome. (**A**) diagnostic odds ratio of percent constriction of pupillary light reflex in predicting a neurological outcome; (**B**) SROC curve of percent constriction of pupillary light reflex in predicting a neurological outcome. Abbreviations: CI, confidence interval; SD, standard deviation; SROC, summary receiver-operating characteristic.

**Figure 5 medicina-58-00804-f005:**
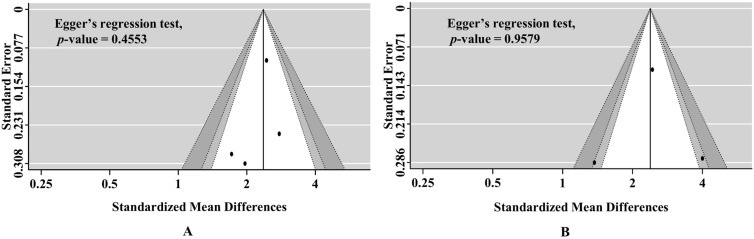
Funnel plot and Egger’s regression test to assess for publication bias. (**A**) Publication bias for percent constriction of pupillary light reflex at 0–24 h after hospital admission and neurological outcome. (**B**) Publication bias for percent constriction of pupillary light reflex at 24–48 h after hospital admission and neurological outcome.

**Table 1 medicina-58-00804-t001:** Characteristics of included studies.

Authors (Year)	Study Design	Country	Sample Size, n	CA Type	GNO, n (%)	Quantitative PLR Assessment Time	Neurological Outcome (Timepoint)
Suys (2014)	sPOS	Switzerland	50	All OHCA	23(46)	0–24 h and 24–48 h	3 months
Heimburger (2016)	sPOS	France	82	OHCA + IHCA	27(32.9)	0–24 h and 24–48 h	3 months
Oddo (2018)	mPOS	Switzerland	434	OHCA + IHCA	181(41)	0–24 h, 24–48 h and 48–72 h	3 months
Riker (2019)	sPOS	USA	55	OHCA + IHCA	16(31)	0–24 h	Hospital discharge

Abbreviations: CA, cardiac arrest; GNO, good neurologic outcome; PLR, pupillary light reflex; sPOS, single-center prospective observational study; mPOS, multicenter prospective observational study; OHCA, out-of-hospital cardiac arrest; IHCA, in-hospital cardiac arrest; h, hour.

## Data Availability

The datasets generated during the current study are available from the corresponding author on reasonable request.

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
