# Peer review of "Efficacy of Quantitative Pupillary Light Reflex for Predicting Neurological Outcomes in Patients Treated with Targeted Temperature Management after Cardiac Arrest: A Systematic Review and Meta-Analysis"

_medicina, 2022, doi:10.3390/medicina58060804_

Round 1
Reviewer 1 Report
Interesting work on a debated topic. The intention to conduct a meta-analysis on such heterogeneous works is ambitious. However, limits have been well underlined and bring to the attention the need to standardize research in this field. It might be relevant to suggest in the conclusions that, in addition to long-term outcomes with larger samples, specific analyzes should be conducted on the influence of sedation and temperature achieved during TTM on the measurements and on outcome prediction.
Reviewer 2 Report
The paper is a good quality review on a medical problem that is already largely traced, that of the accuracy of clinical scores that assess the prognosis of brain damages after cardiac arrest.
An editorial on the subject was published in the Resuscitation Journal the previous month - ,,Novel pupillary assessment in post anoxic coma", especially following the neuronal populations affected by hypoxia and the relationship with pupillary assessment, therefore, qualitative aspects, not quantitative ones
Although the lack of novelty is not necessarily a weakness, the fact that a series of research constantly emphasizes that the relationship between this surveillance criterion and brain recovery is weak, may be a reason to address other issues of interest in the field.
Author Response
Please see the attachment

This manuscript is a resubmission of an earlier submission. The following is a list of the peer review reports and author responses from that submission.